# Extracellular Vesicles from Airway Secretions: New Insights in Lung Diseases

**DOI:** 10.3390/ijms22020583

**Published:** 2021-01-08

**Authors:** Laura Pastor, Elisabeth Vera, Jose M. Marin, David Sanz-Rubio

**Affiliations:** 1Translational Research Unit, Instituto de Investigación Sanitaria de Aragón (IISAragón), Hospital Universitario Miguel Servet, 50009 Zaragoza, Spain; lpastorbernad@gmail.com (L.P.); eliverasolsona@hotmail.com (E.V.); jmmarint@unizar.es (J.M.M.); 2Respiratory Service, Hospital Universitario Miguel Servet, University of Zaragoza, 50009 Zaragoza, Spain; 3Centro de Investigación Biomédica en Red de Enfermedades Respiratorias (CIBERes), 28029 Madrid, Spain

**Keywords:** extracellular vesicle, COPD, asthma, fibrosis, induced sputum, nasal lavage, oral lavage, bronchoalveolar lavage, exosome

## Abstract

Lung diseases (LD) are one of the most common causes of death worldwide. Although it is known that chronic airway inflammation and excessive tissue repair are processes associated with LD such as asthma, chronic obstructive pulmonary disease (COPD) or idiopathic pulmonary fibrosis (IPF), their specific pathways remain unclear. Extracellular vesicles (EVs) are heterogeneous nanoscale membrane vesicles with an important role in cell-to-cell communication. EVs are present in general biofluids as plasma or urine but also in secretions of the airway as bronchoalveolar lavage fluid (BALF), induced sputum (IS), nasal lavage (NL) or pharyngeal lavage. Alterations of airway EV cargo could be crucial for understanding LD. Airway EVs have shown a role in the pathogenesis of some LD such as eosinophil increase in asthma, the promotion of lung cancer in vitro models in COPD and as biomarkers to distinguishing IPF in patients with diffuse lung diseases. In addition, they also have a promising future as therapeutics for LD. In this review, we focus on the importance of airway secretions in LD, the pivotal role of EVs from those secretions on their pathophysiology and their potential for biomarker discovery.

## 1. Introduction

Lung diseases (LD) are among the most common causes of death and disability worldwide [1]. According to the last updates of the Forum of International Respiratory Societies (FIRS), the high incidence of those pathologies is mainly due to global exposure to polluted air, particles, chemicals and infectious microorganisms [1]. Repeated exposures to these agents induce cell damage and trigger immune responses in the airway. Airway epithelium, which is the first line of defense against environmental agents, is damaged by this contact. Interactions and homeostasis between airway epithelium and mesenchymal, endothelial and immune cells in the airway and lungs are also modified [2,3]. Chronic airway inflammation and excessive tissue repair processes result in the development of LD such as asthma, chronic obstructive pulmonary disease (COPD), idiopathic pulmonary fibrosis (IPF) or lung cancer, as well as in systemic inflammation and associated co-morbidities [3,4,5].

Extracellular vesicles (EVs) are heterogeneous nanoscale membrane vesicles, including exosomes, microvesicles and apoptotic bodies, that are secreted by most cell types and they have been mainly described in plasma and serum [6,7,8,9] or urine. However, the presence of EVs has been also reported in secretions from airway such as saliva [10], induced sputum (IS) [11] or bronchoalveolar lavage fluid (BALF) [12]. EVs have emerged as a novel and crucial mechanism of intercellular communication [13,14] with an important role in inflammatory pathologies, including LD. Airway injury modifies EV cargo in lung epithelium derived-EVs promoting an immune response via macrophage activation [15,16]. Therefore, EVs could contribute to clarify the molecular mechanisms underlying LD, but also they could provide new biomarkers among their cargo, as well as new EV-based therapeutic approaches [17,18].

In this review, we aim to summarize current knowledge about EVs related to LD. We will focus on the main LD such as COPD, asthma and IPF, and on the potential of airway secretions as source of new biomarkers. Finally, we suggest the potential function of EVs from these secretions in the pathophysiology of LD and their promising future as origin of biomarkers and for LD.

## 2. Lung Diseases

### 2.1. Asthma

Asthma is one of the most prevalent chronic disease in developed countries in adults and children globally [1], affecting more than 339 million people [19] and 14% of children over the world [1]. Asthma is defined by the international Global Initiative for Asthma (GINA) as a heterogeneous disease related to chronic airway inflammation and characterized by intermittent episodes of wheeze, cough, dyspnea and chest tightness, as well as reversible airway limitation [19]. This disease shows different phenotypes and endotypes, depending on its mechanisms and responses to therapy [20].

Inflammation in asthma is extended from upper to peripheral airway revealing different molecular patterns. Most of the patients suffer allergic asthma characterized by Th2 immune response against environmental stimuli [21]. After antigen presentation, which is driven by dendritic cells [22], T-CD4^+^ lymphocytes secrete Th2-cytokines (IL-4, IL-13 and IL-5). IL-4 and IL-13 stimulate IgE production in B-cells [23], while IL-5 promotes eosinophil maturation in bone marrow and further eosinophil recruitment in the airway [21,24]. Non-allergic T2-eosinophilic response is pushed by alarmins and innate lymphoid cells 2, which are producers of Th2-cytokines, so they trigger eosinophilic inflammation in airway [25]. Finally, non-eosinophilic phenotype is normally related to more severe status and it is generally characterized by neutrophilic airway inflammation [25,26]. Although it is known that non-eosinophilic asthma pathophysiology is driven by Th17/Th1 cells [25] and cytokines as IL-17 and IL-22 [26], its molecular mechanisms are poorly understood. Airway epithelial injury produced by this inflammatory situation triggers asthma-characteristic airway remodeling, including the increase of airway smooth muscle bulk and mucus hypersecretion. Altogether, as shown in Figure 1, these phenomena produce airway narrowing and airway hyperresponsiveness to some environmental agents, which is also a characteristic feature of asthma [27,28].

### 2.2. Chronic Obstructive Pulmonary Disease

COPD has a noteworthy impact on global health, affecting more than 200 million people worldwide [1], leading mortality (3rd cause of death) and morbidity in the world [1,29]. COPD is a LD characterized by persistent airflow limitation and respiratory symptoms due to airway and/or alveolar damage. It is usually caused by significant exposure to irritants, especially cigarette smoke, although sometimes is influenced by host factors as abnormal lung development or α_1_-antitripsin genetic deficiency [30]. COPD is associated to systemic inflammation, which is a risk factor to develop co-morbidities such as atherosclerosis and other cardiovascular diseases, diabetes, metabolic diseases or cancer [31,32].

In contrast to asthma, inflammation in COPD is limited to peripheral airways and lung parenchyma, resulting in fibrosis, tissue destruction, small airway obstruction and emphysema [33]. These mechanisms are irreversible and produce progressive decline in lung function [5]. Different cells are involved in COPD inflammation processes: adaptative immune cells, innate immune cells (mainly neutrophils and macrophages) and structural cells. In response to cigarette smoke or other noxious particles, airway epithelial cells and macrophages are activated to produce several inflammatory mediators [34]. The transcription factor NF-κβ, activated in alveolar COPD macrophages [35], up-regulates the expression of inflammatory molecules [36]. Neutrophils from peripheral airway release CXCL8, promoting autocrine neutrophil recruitment and several proteinases, which contribute to loss of elasticity and peripheral airway collapse. This processes, together with profibrotic effect of TGF- β released from epithelial cells and macrophages, result in alveolar destruction, fibrosis and irreversible airway narrowing [36] (Figure 1). Lungs of COPD patients also suffer an accelerated ageing produced by an abnormal accumulation of senescent structural cells [37]. Finally, eosinophils have been reported to be increased in airway of some COPD patients, suggesting the existence of a different COPD phenotype whose molecular mechanisms are poorly understood [38].

### 2.3. Idiopathic Pulmonary Fibrosis

IPF is increasing over the time worldwide with an estimated incidence of 2.8–18 cases/100,000 people per year in Europe and North America. Lower incidence is defined in Asia and South America, which may be partially due to under-diagnosis [39]. In addition, IPF is more frequent in men [40]. IPF is a chronic, progressive fibrosing interstitial pneumonia of unknown etiology that usually affects to older adults. Disease progression varies between patients and is unpredictable, but IPF leads to severe pathological processes which end in premature death [41].

Molecular mechanisms of IPF are not fully understood although chronic dysregulation in alveolar epithelial cells homeostasis has been described (Figure 1). IPF biopsies show loss of type 1 alveolar epithelial cells (AEC1s) and reduced abnormal renewal capacity in type 2 (AEC2s) [42,43]. Fibroblast aggrupation are also observed between abnormal AECs [44,45]. In response to these microinjuries, remaining AECs are activated to release fibrogenic growth factors and cytokines, resulting in the recruitment of myofibroblast from different sources [45,46,47]. At this location, myofibroblasts increase secretion of abnormal extracellular matrix components, whose accumulation and deposition leads to fibrosis and gas exchange impairment. It also enhances myofibroblast activation, establishing a positive feedback process [48]. In addition, peripheral airway basal cells act as stem cells to repair damaged alveolar airway with new epithelial cells, known as broncholisation [49,50,51].

## 3. Airway Secretions

Sampling the upper and lower airways is a remarkably helpful tool in clinical practice. Different airway secretions are used and has been lately investigated as a promising source of biomarkers, including bronchoalveolar lavage fluid (BALF), induced sputum (IS), nasal lavage fluid (NL) or pharyngeal lavage fluid (PHAL).

### 3.1. Bronchoalveolar Lavage Fluid

BALF is an established procedure for washing the bronchial tree with saline which allows sampling peripheral airways [52]. It is an invasive method, as it is performed together with a flexible fiberoptic bronchoscopy. BALF contains the cells in the alveolar space which have been in contact with the instilled fluid (alveolar macrophages, CD4^+^ and CD8^+^ lymphocytes, neutrophils, eosinophils and mast cells [53]), airway epithelial cells (contaminating BALF if exceed 5% of total cell count [53]), and different soluble components from the endogenous fluid in the alveoli, named the epithelial lining fluid (ELF) [54]. These components proceed mainly from local cell secretory production [55,56,57], but also from the vascular space in contact with the alveolar space [54]. In clinical practice, BALF is used in diagnosis of respiratory infections [58] and as a part of the diagnosis or in characterization of non-infective LD, both in adults and children [59]. However, its invasive nature has limited its use as a research tool.

### 3.2. Induced Sputum

Sputum induction is a relatively non-invasive method to collect a sample of cells and secretions from lower airways as shown in Figure 2 [60]. Sputum is induced by inhalation of hypertonic or isotonic sterile saline solution and obtained by further direct expectoration [61]. IS is a safer and less invasive method to sample ELF than BALF [62,63]. However, IS contains also saliva and squamous cells from upper airways, which are considered contaminant components and represent different proportions of IS [64,65].

Most of IS research has been dedicated to asthma, although multitude of studies have investigated the utility of IS in other LD, especially COPD and chronic cough [66]. Differential cell count in IS is used in clinical practice to assess asthma phenotypes [67], although reference parameters vary among different studies and populations [68,69,70]. In both asthma [71] and COPD [72], cultures from IS are performed to identify the presence of pathogens and manage antibiotic therapy. Extended biomarkers research has been done in IS. For example, assessment of inflammatory pattern of exacerbations in asthma and COPD by IS cytology allows therapy adjustment and prediction of new exacerbation episodes [73]. IS can work as a minimally invasive BALF surrogate in assessment of low airway status in LD with low airway involvement, although further research is needed in order to optimize protocols and standardize reference parameters. In this way, IS has been proposed and studied as a non-invasive surrogate of BALF in interstitial LD diagnosis [74].

### 3.3. Nasal Lavage Fluid

NL is a non-invasive procedure to obtain cytology from nasal cavities [75] (Figure 2). Different nasal lavage techniques have been described [76,77] consisting on instill sterile saline solution. NL is a mixture of saline and nasal secretions, containing epithelial cells, neutrophils, eosinophils, lymphocytes and their secreted mediators [78].

NL has been applied to assess the inflammatory status of a specific part of upper airways, the nasal cavities. Most NL studies have been focused in rhinitis, as it is a disease characterized by nasal mucosa inflammation and nasal hypersecretion [75,78,79]. Recent studies have shown that NL cytology is a best substitute for IS cytology than blood cell count in identification of inflammatory phenotypes of asthma [80,81]. In addition, despite the weaker relationship between nasal mucosa inflammation and COPD, another work revealed that IL-8 concentration in NL have positive correlation with disease severity and cigarette pack-years in COPD patients [82]. NL has a great potential as a non-invasive substitute of IS in those patients or situations where perform IS could be risked.

### 3.4. Pharyngeal Lavage Fluid

PHAL is a recently developed non-invasive technique to collect cells from the oropharyngeal cavity to assess mucosal inflammation of the upper airway [83]. Figure 2 shows that PHAL contains cells from the oropharyngeal mucosa (a major proportion of squamous epithelial cells, followed by a high percentage of neutrophils and other inflammatory cells [83]) and soluble factors and proteins [84].

Higher PHAL lymphocytes number have been found in the patients with obstructive sleep apnea (OSA) [85,86]. Our group reported also that T-CD4^+^cells number in PHAL correlated with more severe OSA status, as well as PHAL concentration of IL-6 and IL-8 [84]. Thus, PHAL is emerging as a new non-invasive tool to assess upper airway inflammation, which could be particularly helpful in some LD with concomitant upper airway inflammation such as asthma.

## 4. Extracellular Vesicles

EVs are a heterogeneous population of small membranous structures released into the extracellular environment by most cells and participate in intercellular communication [87]. EVs have been classified according to different criteria as their size or biogenesis, differentiating between exosomes, microvesicles (MVs), and apoptotic bodies (ABs) [88,89,90]. Exosomes (30–150 nm) result from a complex biogenesis which involves the endosomal membrane invagination, generating intraluminal vesicles (ILVs) inside the late endosome (MVB), and the fusion between the MVB membrane and the plasma membrane releases exosomes [91,92]. MVs (100 nm to 1000 nm), also named ectosomes or microparticles, are generated directly by outward budding of the plasma membrane [93,94]. ABs (1 to 5 µm) are released undergoing apoptosis, via plasma membrane bubbling or fragmentation due to cell disassembly [95,96].

EVs contain biomolecules from their origin cell, including proteins, lipids, metabolites and different nucleic acids (DNA, mRNA, miRNA and other noncoding RNA) [89,97,98]. Their molecular cargo can be uploaded into the vesicles selectively, mostly in exosomes. This process depends on cell status and type [99]. EV release has been reported in platelets [93], tumor cells [100], dendritic cells [101], T [102] and B [103] cells, eosinophils [104], epithelial cells [15,16], endothelial cells [105] or mesenchymal stem cells (MSCs) [106]. Their accessibility in body fluids (plasma [6,8], serum [9], urine [9], saliva [10], BALF [12] or IS [11]) and the specificity of their cargo associated with pathophysiological conditions [107,108] confer them great potential as biomarkers.

EVs have been broadly studied because of their association with pathologies, including inflammatory diseases [109,110] cardiovascular diseases [111,112], metabolic diseases [113,114], cancer [115,116] or LD [107,117,118]. They have been linked to pro-inflammatory processes, participating in antigen presentation [101,103] or macrophage activation [15], as well as to immunomodulatory processes [106]. Additionally, EVs could have a therapeutic approach as vehicles for drug delivery, given their intrinsic biocompatibility and specific target activity [119]. Promising examples of it are novel exosome-based modality of cancer therapy [120] or nanoparticle-guided asthma therapy [121].

In the last decade, the research on EVs has increased. However, the optimal isolation method remains unclear and most times depends on the downstream analyses [13,122,123,124]. Their characterization requires follow several points that include protein markers, size or morphology. Those are described in the MISEV2018 [87] updating the MISEV2014 [125], showing that the EV field is continuously in development and requires further stu-dies.

## 5. EVs in Lung Pathophysiology

Most lung cells release EVs under physiological and pathological conditions. Bronchial epithelial cells (BECs) and alveolar macrophages are the major sources of pulmonary EVs [126], but also vascular endothelial cells, fibroblasts, MSCs and dendritic cells. All together are involved in lung homeostasis, as well as in LD development.

Transfer of EVs cargo between airway epithelial cells can alter the secretions of their target cells, including mucins [127]. Membrane associated mucins are present in tracheobronchial epithelial cells exosomes surface, suggesting their contribution to innate mucosal defense of airway [128]. Similarly, alveolar macrophages release EVs that contain the suppressor of cytokines signaling (SOCS) 1 and 3. These EVs have immunomodulatory effects in the alveolar epithelial cells and their secretion is diminished by cigarette smoking [129]. Table 1 shows some of the proteins, lipids and miRNAs contained in EVs that have been related with asthma, COPD and IPF.

### 5.1. EVs in Asthma

EVs are involved in several processes of asthma immunity and inflammation. One of the most important is antigen presentation ability of EVs released from dendritic cells [101] and B cells [103,130]. Figure 3 shows that MHC-II and co-stimulatory molecules in the surfaces of these EVs allow interaction with T cells and subsequent differentiation of T cells, proliferation of T-CD4^+^ Th2 cells and secretion of Th2 cytokines [101,130].

Asthma-like inflammation driven by IL-13 enhances exosomes secretion in BECs inducing proliferation and chemotaxis of undifferentiated macrophages in the lungs [126]. Bronchoconstriction mechanical stress also stimulates exosomes secretion from BECs, producing typical subepithelial angiogenesis [146]. Furthermore, mechanical signals regulate miRNA cargo of EVs released from BECs [147]. In contrast, EV immunomodulatory effects has been reported to be diminished in asthma, such as EV-mediated transport of SOCS3 from alveolar macrophages to BECs [148].

Eosinophil exosomes of asthmatic patients differ their capacities from non-asthmatic subjects. They promote eosinophil chemotaxis increasing ROS and nitric oxide (NO) levels, and adhesion, via ICAM-1 and integrin-α_2_ upregulation [149] (Figure 3). These exosomes also promote apoptosis and reduce wound healing capacity in BECs via *TNF* or *CCL26* upregulation [150]. In addition, their angiogenic effect on bronchial smooth muscle cells is mediated by *VEGFA* [150]. Similarly, exosomes from neutrophils enhance neutrophils chemotaxis through their leukotriene B4 cargo [136] and adhesion by ICAM-1 upregulation in lung endothelial cells [151].

Few studies describe the effect of asthma pathophysiology on EV-miRNAs [152]. One of them described the increase of miR-223 and miR-142a on EVs from BALF of allergen-treated mice [138] while other, performed on asthmatic patients, described 24 miRNAs differentially expressed including members of let-7 and miR-200 families and miR-21 [139]. MiR-21 targets the pro-inflammatory Th1 cytokine, IL-12, and downregulate Th2 response, which is characteristic of asthma inflammation [153]. Accordingly, miR-21 is increased in serum of asthmatic patients and inversely correlated with FEV_1_ [154]. Lastly, a set of EV-miRNAs from BALF and miRNAs from lung tissue were altered in allergen-exposed mice and were inversely correlated between both origins [155].

### 5.2. EVs in COPD

EVs may also play a main role in COPD pathophysiology. One of the factors associated with COPD, cigarette smoke (CS), modifies EV cargo from different lung cells. CS induces significant release of CCN1-enriched exosomes from BECs, as well as the cleavage of matrix-associated CCN family protein into a shorter isoform [156], which promotes MMP1 secretion in lung epithelial cells contributing to emphysema development [156,157,158] (Figure 3). Similarly, macrophages exposed to CS release MVs with proteolytic and collagenolytic activity in their surface as shown in Figure 3 [159]. In addition, the increased macrophage-MVs release upregulates several pro-inflammatory and chemoattractant mediators, as ICAM-1, IL-8 and MCP-1, in alveolar epithelial cells [160]. Other studies have focused on EV non-coding RNAs in smokers and COPD patients. MiR-210 is overexpressed in both BEC-EVs and lung tissue samples after CS exposure. miR-210 loaded-EVs from COPD patients showed the induction of lung fibroblasts differentiation into myofibroblasts by silencing the autophagy-related factor *ATG7* [140]. Different profiles of long non-coding RNAs (lncRNAs) were found in circulating exosomes from non-smokers and smokers of different smoking products. EV-lncRNAs alterations found in smokers could lead to pulmonary fibrosis and mitophagy, which are hallmark processes in COPD and IPF. In addition, the targets of some of the differential lncRNAs are involved in pathological processes of COPD, IPF and asthma [161].

Overexpression of exosomal miR-21 from BECs also promotes myofibroblast differentiation targeting hypoxia-inducible factor 1α (HIF-1α) [141]. However, decreased miR-21 was observed in BEC exosomes exposed to CS. This compensatory mechanism in response to CS reduced M2 macrophage polarization and, consequently, epithelial-mesenchymal transition involved in airway remodeling [162]. Finally, CS induce on endothelial cells the production of MVs enriched in let-7d, miR-191, miR-126 and miR-125a, whose major effect in recipient macrophages was the impairment of efferocytosis [163].

Surface proteins identified in circulating endothelial MVs suggest that endothelial MVs have an important regulatory role in the coagulation and inflammation processes in COPD [164]. It has been also reported that EV-mediated transport of α_1_-antitrypsin from endothelial cells to alveolar epithelial cells is impaired by CS exposure [131]. In addition, exosomes from activated neutrophils contain neutrophil elastase (NE) in their surfaces, protecting NE from α_1_-antitrypsin proteolysis, and enabling the degradation of the extracellular matrix via NE^+^ exosomes [132] (Figure 3). These results suggest that endothelial and neutrophil derived-EVs contribute to emphysema development in COPD.

### 5.3. EVs in IPF

EVs implication in IPF remains unclear. Up to now, mostly EV studies related to mechanisms involved in fibrosis development used in vitro or murine models and focused their miRNA cargo. MiR-21 is upregulated in EVs from serum of a lung fibrosis mice model and from human IPF patients and it is correlated with IPF progression and mortality [143]. Another study showed that miR-21 upregulation is promoted via TGF-β1 and it has pro-fibrotic effects including myofibroblast differentiation as described in Figure 3 [165]. Kidney epithelial cells have been reported to release an increased amount of exosomes in response to injury, loaded with specific pro-fibrotic cargo such as TGF-β1 mRNA, which produce fibroblasts proliferation, activation and repair processes [166].

On an IPF rat model, miR-328 was overexpressed in M2 macrophage-derived exosomes with associated proliferative effects in fibroblasts via silencing *FAM13A* and overexpression of certain proteins as collagen 1A and α-SMA (Figure 3) [144]. Three more exosomal microRNAs have been reported to be altered in IS from IPF patients, miR-33a, miR-142 and let-7d [145].

Lastly, a recent study showed increase amount of EVs carrying WNT5A in BALF from IPF patients. These EVs originating from lung fibroblasts have autocrine effects in vitro stimulating fibroblast proliferation [135]. Impairment or block of EV-WNT5A suppresses the fibrogenic effects of EVs improving the IPF pathology.

## 6. EVs in Airway Secretions and Potential Utility in Clinical Practice

Presence of EVs in airway secretions was first described on BALF of healthy subjects [12]. By contrast, the isolation of EVs from sputum was reported 13 years later from spontaneous sputum of cystic fibrosis patients [167]. Following we present the emerging relevance of EVs from airways secretion on LD.

### 6.1. Airway EVs in Asthma

The vast majority of EV studies in patients with asthma have been done using BALF. Increased amount of exosomes with specific protein surface markers [168], lipid [168] and miRNA [139] profiles were found in BALF of asthmatic patients. Exosome concentration and expression of altered EV-miRNAs were correlated with asthma severity biomarkers such as serum eosinophil and IgE levels [168] and FEV_1_ [139]. Up to now, only one work has recovered EVs from IS of asthmatic patients [11]. Exosomes from IS of mild allergic asthma patients contain short RNA species and surface markers as tetraspanins and HLA-DR, suggesting the feasibility of IS as an accessible minimally-invasive source of EV biomarkers [11].

EVs were isolated from NL for the first time on 2011 [169]. More than 600 proteins have been identified among NL-EV cargo, finding proteins related with antimicrobial and barrier functions decreased in patients with asthma and rhinitis [170]. Patients with nasal polyps, commonly associated with asthma, displayed increased levels of ADAM10 in exosomes from NL, which promote angiogenesis and vascular permeability contributing to nasal polyps formation [171].

Asthma is a complex and not fully understood LD. It chronic inflammatory status and remodeling is located in the airway. As we mentioned before, several studies have shown the relationship of EVs from IS, BALF or NL with asthma. Therefore, EVs obtained from the different airway secretions could help to clarify what is happening in the pathology. Those EVs are secreted from the cells involved in the pathogenesis and they have a great potential as both diagnostic tool and therapeutic target. As biomarkers, they can provide a new and more accurate definition of asthma phenotypes, which is crucial for a better treatment. However, it is necessary further analyses that fully characterized both the EVs and their cargo. Specially, of great interest those studies comparing between not only healthy subjects and asthmatic patients but also between different profiles of patients.

In terms of EV-based therapeutics, synthetic liposomes loaded with SOCS3, whose secretion is impaired in asthma, restored the immunomodulatory effects of EVs in an allergic asthma murine model [148]. Finally, MSC-derived EVs have promising immunomodulatory effects in both allergic asthma murine models [172,173] and in vitro cell cultures [173,174]. MSC-derived EVs induce regulatory T cells proliferation on asthmatic patients [174] and inhibit ILC2 activity from allergic rhinitis patients and a murine asthma model [173]. Those results implies their promising future for asthma therapy although extensive studies in vitro and in animal models are required before their translation to the clinic.

### 6.2. Airway EVs in COPD

EVs from airway secretions have also emerged as a potential tool for COPD research, mainly those from BALF. Exosomes from BALF of COPD patients expressed CD66b and NE inducing a COPD phenotype into normal mice [132]. Furthermore, in one study, three EV-miRNAs (let-7e, let-7g and miR-26b) were found altered in BALF of smokers [142]. Furthermore, those smokers BALF-EVs altered human BECs in vitro promoting lung cancer and COPD-like phenotype development [142]. Similarly, 3 exosomal miRNAs from BALF have shown high discriminatory power between COPD patients and healthy ex-smokers [175]. IS-MVs from COPD patients were classified by their cellular origin, being endothelial CD31^+^ MVs and granulocyte CD66b^+^ MVs correlated with COPD severity parameters [133]. Those studies show the importance of airway EVs and their cargo in the progression of COPD, even their ability to promote it phenotype.

COPD is a complex pathology with high impact in the society. Although there is a significant number of studies focus on this pathology, several aspects of it mechanisms and pathophysiology remain unclear. Among them, the heterogeneity of COPD patients is a critical point. Airway EVs, due to their specific cargo, could help to classify the patients depending on their risk to suffer a mainly parenchymal injure, developing an emphysema, and those patients whose main affectation is the bronchial tree, showing chronic bronchitis and the subsequently airway remodeling. Studies that evaluate the cargo of those phenotypes are required for their transalation to the precision medicine.

EVs constitute also a novel therapeutic strategy for COPD. Recently, it has been developed an inhaled formula to target a specific miRNA to lung cells by packaging it into synthetic nanoparticles [176]. Another protocol to transform mice serum derived-EVs with small RNAs were developed and intratracheally administrated in mice with induced lung inflammation. Those EVs were taken up by alveolar macrophages and successfully modulated lung inflammation without trigger immunogenic adverse responses [177]. Those findings set the basis for the development of inhaled EV-based therapies for LD.

### 6.3. Airway EVs in IPF

The interest on the role of EVs in fibrotic diseases have raised during the last years [178]. Two different studies on BALF showed that EVs from IPF patients increase their tissue factor [134] and WNT5A [135] levels. Those tissue factor-bearing MVs were inversely correlated with lung function parameters [134]. In addition, blocking WNT5A reduces the profibrotic effects of those EVs, proposing WNT5A as a potential diagnostic and therapeutic target [135]. In case of IS, a panel of 3 exosomal miRNAs (miR-142-3p, let-7d-5p and miR-33a-5p) allowed to distinguish IPF from healthy subjects. Furthermore, two of them, miR-142-3p and let-7d-5p, were correlated with parameters of IPF severity such as lung diffusion capacity and alveolar-capillary function [145].

Patients with IPF also display a high heterogeneity. While in some of them the disease progresses slowly, others suffer a quick progression with the fatal consequences associated with this pathology. The EVs and their cargo could explain those differences helping to stratify different group of patients and to apply an individually therapy for each subject. Further studies of large population that include different profiles of IPF patients will contribute to those aspects.

EV-based therapeutic approaches for IPF have been barely investigated. EVs from human bone marrow MSCs prevent and alleviate IPF features in a murine model by restoring normal monocyte/macrophage profile, both in vivo and in vitro [179]. By contrast, other studies have focused on antifibrotic effects of specific populations of EVs. Activation of fibroblasts stimulates their EV release carrying PGE_2_, producing inhibition of both TGF-β-induced myofibroblast differentiation and excessive extracellular matrix production on naïve fibroblast [137]. Similarly, MSC-derived EVs also showed to exert aforementioned effects on fibroblasts via their specific miRNA cargo [180]. Although those studies are few, their potential is great due to the absence of an optimal treatment for IPF.

## 7. Conclusions

Asthma, COPD and IPF are multifactorial and complex inflammatory LD which represent an immense health challenging globally. The specific pathways involved in their pathophysiology remains unknown. Airway secretion analysis are included among the routine clinical practice of these diseases; however, their potential is still underestimated despite of their low invasion. EVs could play a crucial role in LD pathogenesis and progression, including asthma, COPD and IPF. Their capacity to transfer information between several cell types involved in LD confers them and their cargo the potential of being not only a novel source of diagnostic and prognostic biomarkers, but also a new therapeutic tool. Despite their potential, some aspects need to be improve before the translation of EV to medical practice. Although the field is evolving fast, the difficulty to stablish an optimal isolation method, the inter-laboratory reproducibility of the results and the high-cost of the equipment are some of these barriers.

The relevance of EVs from airway secretions on LD could be even higher than circulating EVs due to the physiological origin of these diseases. Nevertheless, their technical issues are also more difficult because of the relatively low EV concentration in airway secretions compared to blood. Therefore, further studies are needed to overcome some challenges such as stablish an optimal EVs isolation, fully characterized those EVs or complete reliable profiles of not only EV-miRNAs and EV-proteins but also EV-lncRNA and EV-lipids, specifically in those airway secretions obtained with non-invasive methods as induced sputum or nasal lavage. Additionally, more studies are needed to describe the relationship between the EVs present in different airway secretions, peripheral blood and lung parenchima in the same subject. Therefore, we can clarify the origin and distribution of those EVs and the role they play in the lung and the lower airway. Altogether, those studies may provide novel specific and minimally non-invasive biomarkers, which would surely improve the clinical management of LD.

## Figures and Tables

**Figure 1 ijms-22-00583-f001:**
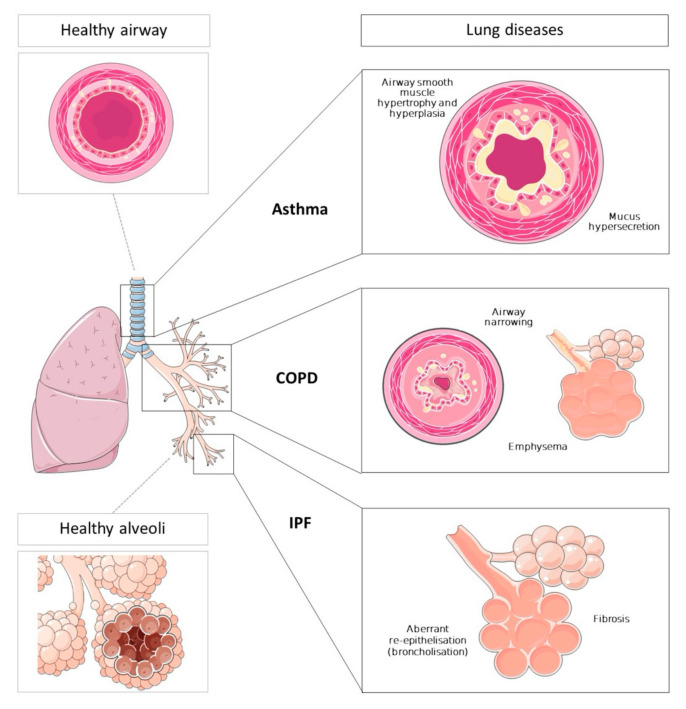
Overview of the main processes and events that occur in different regions of lung and airway in asthma, chronic obstructive pulmonary disease (COPD) and interstitial pulmonary fibrosis (IPF) patients, compared to healthy condition. Asthma involves mainly the upper airway promoting airway remodeling, which includes airway smooth muscle hypertrophy and hyperplasia, and goblet cell hyperplasia and consequent mucus hypersecretion. These processes cause reversible airway narrowing. COPD affects lower airways and lung parenchyma causing irreversible airway narrowing and alveolar disruption (emphysema) in the lungs. IPF affects only the lungs through excessive fibrosis processes with subsequent aberrant alveolar re-epithelisation (broncholisation). (Components of this figure have been obtained and modified from Servier Medical Art, https://smart.servier.com, with permission under the Creative Commons Attribution 3.0 Unported License.).

**Figure 2 ijms-22-00583-f002:**
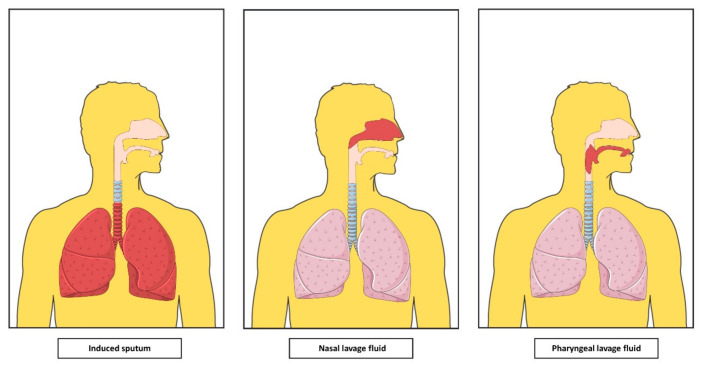
Anatomical origin of minimally invasive and non-invasive airway secretions. Induced sputum comes from lower airway while nasal lavage and pharyngeal lavage have their origin in different parts of the upper airway. (Components of this figure have been obtained and modified from Servier Medical Art, https://smart.servier.com, with permission under the Creative Commons Attribution 3.0 Unported License.).

**Figure 3 ijms-22-00583-f003:**
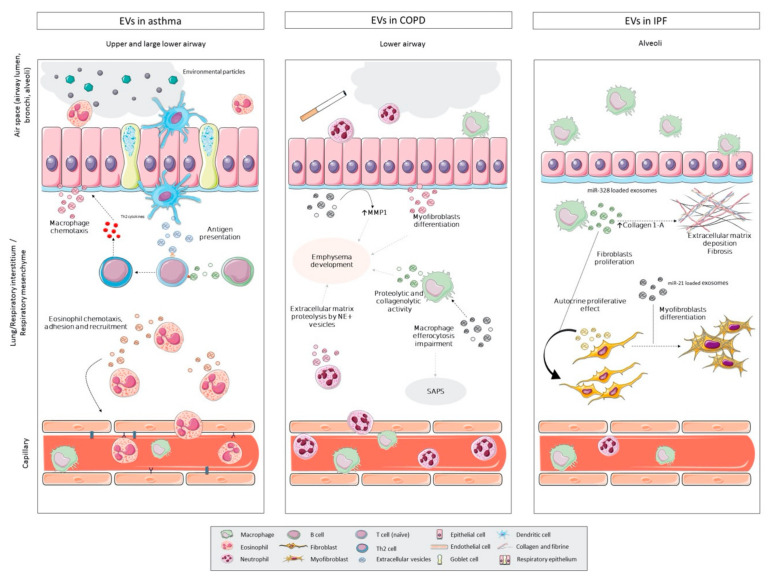
Summary of some of the main reported roles of extracellular vesicles (EVs) in the pathophysiological mechanisms of asthma, chronic obstructive pulmonary disease (COPD) and interstitial pulmonary fibrosis (IPF). The figure shows the cellular origin of the EVs as well as their cellular destiny or their main action. In the upper and large lower airway of asthma patients, EVs contribute to antigen presentation, EVs from endothelial cells participate in macrophage chemotaxis while eosinophil EVs could act in their own chemotaxis, adhesion and recruitment. In COPD, EVs could promote emphysema development increasing MMP1 secretion in lung epithelial cells, through myofibroblast differentiation or enhancing the proteolysis of the extracellular matrix. In the alveoli of IPF patients, EVs from macrophages could increase the collagen 1-A expression in fibroblast, increasing the extracellular matrix deposition, as well as promote fibroblast proliferation. (Components of this figure have been obtained and modified from Servier Medical Art, https://smart.servier.com, with permission under the Creative Commons Attribution 3.0 Unported License.).

**Table 1 ijms-22-00583-t001:** Proteins, lipids and miRNAs contained in extracellular vesicles (EVs) that have been associated with asthma, chronic obstructive pulmonary disease (COPD) and idiopathic pulmonary fibrosis (IPF).

EV Cargo	Asthma	COPD	IPF
Proteins	ADAM10 [11], MHC-II [101,103,130], HLA-DR (IS) [11]	α1-antitrypsin [131], CD66b [132,133], CD31, NE [132]	Tissue factor [134], WNT5A [135]
Lipids	Leukotriene B4 [136]		PGE2 [137]
miRNAs	miR-223 and miR-142a [138], let-7, miR-200 families, miR-21 [139]	miR-210 [140], miR-21 [141], Let-7e, let-7g, miR-26b [142]	miR-21 [143], miR-328 [144], miR-33a, miR-142 and let-7d [145]

## Data Availability

Not applicable.

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
