# Peer review of "Extracellular Vesicles from Airway Secretions: New Insights in Lung Diseases"

_ijms, 2021, doi:10.3390/ijms22020583_

Round 1

Reviewer 1 Report

In this review article by Pastor et al. titled “Extracellular Vesicles from Airway Secretions: New Insights in Lung Diseases”, the authors give an overview of the different methods of sample collection that could be used for the analysis of extracellular vesicles [EVs] as well as discuss some recent findings regarding the role of EVs in three major respiratory diseases (i.e., asthma, chronic obstructive pulmonary disease [COPD], and idiopathic pulmonary fibrosis [IPF]). While the role of EVs in both health and in disease is a timely topic, I have some concerns regarding the manuscript as I detail below.

Major comments:

1, Overall, I felt the manuscript could be better organized. The authors independently give an overview of the respiratory diseases in question, the various airway secretions (or more precisely, the method of collection), the different classes of EVs, before finally reaching the topic to be discussed in this review: the role of EVs in lung disease. The discussion regarding the role of EVs in respiratory diseases (sections 5 and 6) seemed much weaker compared to the amount of text allocated to background information (sections 1-4).

2, The authors need to strengthen the discussion to provide an original contribution to the topic. The current manuscript does give an overview of recent studies, but little is discussed about current gaps in knowledge and how the field may overcome these gaps. For example, one part of the manuscript that could be improved as such is the discussion presented in lines 389-392: what kind of future studies? What are the current obstacles, and how can the obstacles be overcome?

3, A friendly suggestion regarding the abstract: The examples referred to in the abstract are somewhat too detailed for the abstract. I think a more general rationale for why this topic (EVs in lung disease) matters should be presented.

4, Regarding the figures, I felt that the main message each figure was trying to convey was unclear especially in relationship to the discussion presented in the main text. The figure legends should be more detailed, and it should unambiguously indicate what the figure shows.

Minor comments:

1, Line 18, “Modifications of airway EV cargo”: If the authors meant disease-specific changes in airway EV cargo, I feel that “alterations” is a better term than “modification”, as the latter term connotes a more subjectively induced change.

2, Line 29: Please add a reference.

3, Line 65, “citoquines”: Did the authors wish to say cytokines?

4, Line 206, “EVs contain different biomolecules from their origin cell, including proteins, lipids, metabolites 206 and different nucleic acids (DNA, mRNA, miRNA and other noncoding RNA)”: By “different” did the authors mean 1) different composition as compared to the cell-of-origin, 2) molecules of various classes or 3) something else? The wording of this sentence should be modified to make the point more clear.

Author Response

In this review article by Pastor et al. titled “Extracellular Vesicles from Airway Secretions: New Insights in Lung Diseases”, the authors give an overview of the different methods of sample collection that could be used for the analysis of extracellular vesicles [EVs] as well as discuss some recent findings regarding the role of EVs in three major respiratory diseases (i.e., asthma, chronic obstructive pulmonary disease [COPD], and idiopathic pulmonary fibrosis [IPF]). While the role of EVs in both health and in disease is a timely topic, I have some concerns regarding the manuscript as I detail below.

We thank Reviewer #1 very much for his/her insightful comments. We have revised our manuscript according to the reviewers´ suggestions. A point by point answer is below:

 Major comments:

  • Overall, I felt the manuscript could be better organized. The authors independently give an overview of the respiratory diseases in question, the various airway secretions (or more precisely, the method of collection), the different classes of EVs, before finally reaching the topic to be discussed in this review: the role of EVs in lung disease. The discussion regarding the role of EVs in respiratory diseases (sections 5 and 6) seemed much weaker compared to the amount of text allocated to background information (sections 1-4)

We thank the Reviewer for his/her assessment. We agree with the Reviewer that sections 5 and 6 are the most important in the text and following this indication, we have now extended this part with some discussion. Regarding the structure, in the review we aim to show the state of the art of EVs in the main lung diseases and the potential of EVs contained in airway secretions in those pathologies. The first section tried to clarify the importance of the chosen lung diseases and their main characteristics. After that, we introduce the different type of airway secretions and their potential clinical uses in LD patient management. Then, we provide a short overview of the EVs before going in depth with the role of EVs in pathogenesis of LD and their potential future role as biomarkers of clinical outcomes in patients with LD. As we mentioned before we added a final paragraph for each pathology concerning this airway EV  discussing their potential and limitations to strengthen this part of the review. We finish our work with a conclusion section that contains the main aspects exposed in the previous parts and the final message of our review.  

  • The authors need to strengthen the discussion to provide an original contribution to the topic. The current manuscript does give an overview of recent studies, but little is discussed about current gaps in knowledge and how the field may overcome these gaps. For example, one part of the manuscript that could be improved as such is the discussion presented in lines 389-392: what kind of future studies? What are the current obstacles, and how can the obstacles be overcome?

Thank you for this excellent observation. According with your suggestions, we have include an additional discussion paragraph about the potential use of EVs obtained from airway secretions as new research tool in each LD. The three paragraphs added are the followed:

Lines 382-391: “Asthma is a complex and not fully understood LD. It chronic inflammatory status and remodeling is located in the airway. As we mentioned before, several studies have shown the relationship of EVs from IS, BALF or NL with asthma. Therefore, EVs obtained from the different airway secretions could help to clarify what is happening in the pathology. Those EVs are secreted from the cells involved in the pathogenesis and they have a great potential as both diagnostic tool and therapeutic target. As biomarkers, they can provide a new and more accurate definition of asthma phenotypes, which is crucial for a better treatment. However, it is necessary further analyses that fully characterized both the EVs and their cargo. Specially, of great interest those studies comparing between not only healthy subjects and asthmatic patients but also between different profiles of patients.”

Lines 397-399: “Those results implies their promising future for asthma therapy although extensive studies in vitro and in animal models are required before their translation to the clinic.

Lines 409-417: “Those studies show the importance of airway EVs and their cargo in the progression of COPD, even their ability to promote it phenotype.

COPD is a complex pathology with high impact in the society. Although there is a significant number of studies focus on this pathology, several aspects of it mechanisms and pathophysiology remain unclear. Among them, the heterogeneity of COPD patients is a critical point. Airway EVs, due to their specific cargo, could help to classify the patients depending on their risk to suffer a mainly parenchymal injure, developing an emphysema, and those patients whose main affectation is the bronchial tree, showing chronic bronchitis and the subsequently airway remodeling. Studies that evaluate the cargo of those phenotypes are required for their transalation to the precision medicine.

Lines 434-438: “Patients with IPF also display a high heterogeneity. While in some of them the disease progresses slowly, others suffer a quick progression with the fatal consequences associated with this pathology. The EVs and their cargo could explain those differences helping to stratify different group of patients and to apply an individually therapy for each subject. Further studies of large population that include different profiles of IPF patients will contribute to those aspects.

Lines 445-446: “Although those studies are few, their potential is great due to the absence of an optimal treatment for IPF.

Additionally, we have also modified the conclusion section as the Reviewer suggested to clarify the final message and the future of airway EVs in LD.

Lines 455-470: “Despite their potential, some aspects need to be improve before the translation of EV to medical practice. Although the field is evolving fast, the difficulty to stablish an optimal isolation method, the inter-laboratory reproducibility of the results and the high-cost of the equipment are some of these barriers.

The relevance of EVs from airway secretions on LD could be even higher than circulating EVs due to the physiological origin of these diseases. Nevertheless, their technical issues are also more difficult because of the relatively low EV concentration in airway secretions compared to blood. Therefore, further studies are needed to overcome some challenges such as stablish an optimal EVs isolation, fully characterized those EVs or complete reliable profiles of not only EV-miRNAs and EV-proteins but also EV-lncRNA and EV-lipids, specifically in those airway secretions obtained with non-invasive methods as induced sputum  or nasal lavage. Additionally, more studies are needed to describe the relationship between the EVs present in different airway secretions, peripheral blood and lung parenchima in the same subject. Therefore, we can clarify the origin and distribution of those EVs and the role they play in the lung and the lower airway. Altogether, those studies may provide novel specific and minimally non-invasive biomarkers, which would surely improve the clinical management of LD.”

  • A friendly suggestion regarding the abstract: The examples referred to in the abstract are somewhat too detailed for the abstract. I think a more general rationale for why this topic (EVs in lung disease) matters should be presented.

We are grateful for this comment. Following the indications mentioned above, we have removed from the abstract the part that detailed:  “For example, EVs from NL displayed decreased levels of proteins related with antimicrobial and barrier functions in patients with asthma. EVs from BALF of COPD patients were able to induce a COPD phenotype into normal mice. A profile of EV-miRNAs from IS distinguished between patient with IPF and healthy subjects.”

We have replaced this part as follow: “Airway EVs have shown a role in the pathogenesis of some LD such as eosinophil increase in asthma, the promotion of lung cancer in vitro models in COPD and as biomarkers to distinguishing IPF in patients with diffuse lung diseases. In addition, they also have a promising future as therapeutics for LD.”

  • Regarding the figures, I felt that the main message each figure was trying to convey was unclear especially in relationship to the discussion presented in the main text. The figure legends should be more detailed, and it should unambiguously indicate what the figure shows.

Thank you for pointing this out. We agree with the Reviewer that the figure legend could be much clear and therefore we have modified the figure legends of the three figures. The new modified figure captions are as follow:

“Figure 1. Overview of the main processes and events that occur in different regions of lung and airway in asthma, chronic obstructive pulmonary disease (COPD) and interstitial pulmonary fibrosis (IPF) patients, compared to healthy condition. Asthma involve mainly the upper airway promoting airway remodeling, which includes airway smooth muscle hypertrophy and hyperplasia, and goblet cell hyperplasia and consequent mucus hypersecretion. These processes cause reversible airway narrowing. COPD affects lower airways and lung parenchyma causing irreversible airway narrowing and alveolar disruption (emphysema) in the lungs. IPF affects only the lungs through excessive fibrosis processes with subsequent aberrant alveolar re-epithelisation (broncholisation). [Components of this figure have been obtained and modified from Servier Medical Art, https://smart.servier.com, with permission under the Creative Commons Attribution 3.0 Unported License.].”

“Figure 2. Anatomical origin of minimally invasive and non-invasive airway secretions. Induced sputum comes from lower airway while nasal lavage and pharyngeal lavage have their origin in different parts of the upper airway. [Components of this figure have been obtained and modified from Servier Medical Art, https://smart.servier.com, with permission under the Creative Commons Attribution 3.0 Unported License.].”

“Figure 3. Summary of some of the main reported roles of extracellular vesicles (EVs) in the pathophysiological mechanisms of asthma, chronic obstructive pulmonary disease (COPD) and interstitial pulmonary fibrosis (IPF). The figure shows the cellular origin of the EVs as well as their cellular destiny or their main action. In the upper and large lower airway of asthma patients, EVs contribute to antigen presentation, EVs from endothelial cells participate in macrophage chemotaxis while eosinophil EVs could act in their own chemotaxis, adhesion and recruitment. In COPD, EVs could promote emphysema development increasing MMP1 secretion in lung epithelial cells, through myofibroblast differentiation or enhancing the proteolysis of the extracellular matrix. In the alveoli of IPF patients, EVs from macrophages could increase the collagen 1-A expression in fibroblast, increasing the extracellular matrix deposition, as well as promote fibroblast proliferation.  [Components of this figure have been obtained and modified from Servier Medical Art, https://smart.servier.com, with permission under the Creative Commons Attribution 3.0 Unported License.].”

Regarding their relation with the main text, the Figure 1 try to show the different part of respiratory system and the main processes affected by the LD described in this Review. The Figure 2 try to indicate the origin (different parts of the airway) of the different airway secretions. Finally, the Figure 3 try to show the different mechanisms in which the EVs could be involved in the LD described based in the studies referred in the main text. 

Minor comments:

  1. Line 18, “Modifications of airway EV cargo”: If the authors meant disease-specific changes in airway EV cargo, I feel that “alterations” is a better term than “modification”, as the latter term connotes a more subjectively induced change.

We thank for this comment and fully agree with the Reviewer. We have changed the term “Modifications” by “Alterations”.

  1. Line 29: Please add a reference.

We have added the corresponding reference at this point.

  1. Line 65, “citoquines”: Did the authors wish to say cytokines?

We apologise for our mistake and thanks the Reviewer for this correction. We have solved this issue in the line 65. We additionally double-checked the main text to avoid this failure.

  1. Line 206, “EVs contain different biomolecules from their origin cell, including proteins, lipids, metabolites 206 and different nucleic acids (DNA, mRNA, miRNA and other noncoding RNA)”: By “different” did the authors mean 1) different composition as compared to the cell-of-origin, 2) molecules of various classes or 3) something else? The wording of this sentence should be modified to make the point more clear.

According to the Reviewer comment, we have deleted the word different to avoid misunderstandings. We pretended to show that among EV cargo there are different types of biomolecules such as protein, lipids or RNA.

We much appreciate the reviewer’s comments and suggestions. We have addressed fully all reviewer’s comments and suggestions that have greatly improved our manuscript.

Reviewer 2 Report

Pastor ET AL present a paper entitled "Extracellular Vesicles from Airway Secretions: New Insights in Lung Diseases". 

The review is very interesting and reports a well organized summary on the EV in lung diseases. The figures are beaitiful.

However, I have some comments that could improve the paper:

-the authors should add a table to summarize the main proteins and miRNAs associated with EVs. Otherwise, the paper is difficult to follow, whereas a table would help the reader.

-what about the other types of RNA enclosed in EVs (i.e long non coding RNA, circular RNA etc). the lncRNAs are becoming very important and i think that the authors should add information about that

Author Response

Pastor ET AL present a paper entitled "Extracellular Vesicles from Airway Secretions: New Insights in Lung Diseases". 

The review is very interesting and reports a well organized summary on the EV in lung diseases. The figures are beaitiful.

We thank Reviewer #2 very much for his/her insightful comments. We have revised our manuscript according to the reviewers´ suggestions. A point by point answer is below:

However, I have some comments that could improve the paper:

-the authors should add a table to summarize the main proteins and miRNAs associated with EVs. Otherwise, the paper is difficult to follow, whereas a table would help the reader.

We are grateful for this comment. Following the Reviewer suggestion, we have include a new Table in the main text, Table 1. In this table we include the proteins, lipids and miRNAs from EVs that are related with the pathologies described, asthma, COPD and IPF. We have introduce de table in the text at line 256-257 as follow “Table 1 shows some of the proteins, lipids and miRNAs contained in EVs that have been related with asthma, COPD and IPF.”

Next, we show both Table caption and Table 1:

Table 1. Proteins, lipids and miRNAs contained in EVs that have been associated with asthma, chronic obstructive pulmonary disease (COPD) and idiopathic pulmonary fibrosis (IPF).

EV Cargo

Asthma

COPD

IPF

Proteins

ADAM10 [11], MHC-II [101, 103, 130], HLA-DR (IS) [11]

α1-antitrypsin [131], CD66b [132, 133], CD31, NE [132]

Tissue factor [134], WNT5A [135]

Lipids

Leukotriene B4 [136]

PGE2 [137]

miRNAs

miR-223 and miR-142a [138], let-7, miR-200 families, miR-21 [139]

miR-210 [140], miR-21 [141], Let-7e, let-7g, miR-26b [142]

miR-21 [143], miR-328 [144], miR-33a, miR-142 and let-7d [145]

Finally, we have modified and checked the reference section to be consistent with the changes introduced in the Table 1.

-what about the other types of RNA enclosed in EVs (i.e long non coding RNA, circular RNA etc). the lncRNAs are becoming very important and i think that the authors should add information about that

Thank you for pointing this out. We agree that lncRNA enclosed in EVs could have also a major role in a near future. However, we have only found one study that relate this type of non-coding RNA in EVs with asthma, COPD or IPF. Following your recommendation, we added this work in the main text at lines 301-305: “Different profiles of long non-coding RNAs (lncRNAs) were found in circulating exosomes from non-smokers and smokers of different smoking products. EV-lncRNAs alterations found in smokers could lead to pulmonary fibrosis and mitophagy, which are hallmark processes in COPD and IPF. In addition, the targets of some of the differential lncRNAs are involved in pathological processes of COPD, IPF and asthma [161].”

Finally, we think that it is really interesting to further investigate in this lncRNA so we also included a reference to this type of ncRNA in the conclusion (line 464): “but also EV-lncRNA and EV-lipids”.

We much appreciate the reviewer’s comments and suggestions. We have addressed fully all reviewer’s comments and suggestions that have greatly improved our manuscript.

Round 2

Reviewer 1 Report

Thank you for addressing the comments I have raised in my previous report. I felt that the manuscript has been improved, and the discussion is strengthened compared to the previous version. However, I would like to suggest English proofreading—especially of the revised portions—as it contains grammatical/typographical errors.

Reviewer 2 Report

the authors have responded to my comments and suggestions